# Threshold Segmentation and Length Measurement Algorithms for Irregular Curves in Complex Backgrounds

**DOI:** 10.3390/s22155761

**Published:** 2022-08-02

**Authors:** Xusheng Ruan, Honggui Deng, Qiguo Xu, Yang Liu, Jun He

**Affiliations:** School of Physics and Electronics, Central South University, Lushan South Road, Changsha 410083, China; csu_ruan@csu.edu.cn (X.R.); 202211045@csu.edu.cn (Q.X.); liuyang999@csu.edu.cn (Y.L.); junhe@csu.edu.cn (J.H.)

**Keywords:** irregular curves, quasi-bimodal threshold segmentation, single-pixel skeleton, length measurement

## Abstract

It is an urgent problem to know how to quickly and accurately measure the length of irregular curves in complex background images. To solve the problem, we first proposed a quasi-bimodal threshold segmentation (QBTS) algorithm, which transforms the multimodal histogram into a quasi-bimodal histogram to achieve a faster and more accurate segmentation of the target curve. Then, we proposed a single-pixel skeleton length measurement (SPSLM) algorithm based on the 8-neighborhood model, which used the 8-neighborhood feature to measure the length for the first time, and achieved a more accurate measurement of the curve length. Finally, the two algorithms were tested and analyzed in terms of accuracy and speed on the two original datasets of this paper. The experimental results show that the algorithms proposed in this paper can quickly and accurately segment the target curve from the neon design rendering with complex background interference and measure its length.

## 1. Introduction

As cities develop, energy-saving and environmentally friendly neon lights have become a meaningful way to enhance the image of a city [1,2] and an essential part of the urban night scene [3]. Although the patterns vary in style, the vital elements are all irregular curves. Measuring the length of the distinctive curves that make up the pattern from the neon design renderings is necessary before production. It significantly improves efficiency, saves raw materials, and guides production. In addition, in the field of construction work, measuring and analyzing the number and length of cracks on building surfaces based on captured pictures of bridges, tunnels, roads, and other facilities is a significant way to assess their risk and quality [4,5]. Since there are various background interferences in addition to the target curve in the design drawings and captured pictures, it is of great significance in engineering practice to measure the length of the curves in the images with background interference.

Since separating the background noise from the binarized image is difficult, it is necessary to remove the noise interference before the length measurement. The traditional segmentation method is manual tracing, which has low measurement efficiency and significant error in the results. The blue light and ultraviolet rays from the computer screen will cause damage to the staff’s eyes [6,7]. With the development of computer technology, image segmentation technology to segment and extract target curves in design drawings has gradually become a better way to replace manual tracing. Threshold segmentation is a technique with the most straightforward principle and the most comprehensive application range in image segmentation technology. Two classical threshold segmentation algorithms, the bimodal method [8] and OTSU [9], directly perform image segmentation according to the grayscale difference between the target and the background. The principle is simple, but it only considers the target segmentation under a single background, and the segmentation effect on images with complex backgrounds is poor. AL-Smadi et al. proposed a foreground bimodal segmentation algorithm based on the bimodal algorithm. The algorithm can accurately segment images in urban traffic scenes, but the operation steps are cumbersome, and the algorithm’s complexity is very high [10]. Researchers combined the classical OTSU algorithm with the grayscale histogram and proposed a new unsupervised segmentation algorithm. The algorithm runs quickly, but the main application scenario is the rough estimation of the target, and the segmentation accuracy is low [11]. In addition, some researchers improve the OTSU algorithm by modifying the weight factor [12,13]. Compared with the foreground bimodal segmentation algorithm, these improved methods maintain a lower time complexity and improve the segmentation accuracy to a certain extent. However, it is only suitable for images with less background interference, and the accuracy of target segmentation is low in images with more complex backgrounds. The algorithms in [11,12,13] all belong to improved OTSU segmentation algorithms, which are relatively common in complexity compared to the algorithm in [10]. Still, none of them solve the problem of low accuracy when extracting targets from complex backgrounds.

After getting the target curve, the next step is to measure the length. There are two ways to measure the length of irregular curves: direct measurement and indirect measurement. Direct measurements generally begin with the refinement of the curve to obtain a single-pixel skeleton. Then the line length is measured by measuring the size of the single-pixel structure. Many researchers have investigated direct measurement methods. Kim et al. proposed using the coordinate difference between the two endpoints of the crack skeleton in the X or Y direction as the measured value when measuring the length of the concrete fracture area. This method is suitable for a rough estimate of extent, but cannot accurately measure fracture length [14]. Then, after obtaining the single-pixel skeleton of the target curve, the researchers first received the total number of pixels on the structure by counting or integrating, and then multiplied the length of a single pixel to calculate the size of the curve [15,16,17]. This method is simple in logic and easy to implement, but oversimplifies the problem, resulting in low calculation accuracy. To further improve the accuracy, the researchers improved the method which was used in the literature [15,16,17] by enhancing the integrand [18], introducing the idea of displacement [19,20], and classifying the pixels [21,22]. Although the angle of improvement is different, these methods essentially use “1” and “$\sqrt{2}$” to replace the length represented by a pixel and calculate the total distance by accumulating the sizes of all pixels on the skeleton. These methods embody the idea of classification and improve measurement accuracy to a certain extent. However, the measurement accuracy still needs further improvement. In addition to directly measuring the length of the curve through the single-pixel skeleton, researchers also use indirect measurement to measure the length. Some researchers use image thinning to extract the edges of the original curve and use half the length as a measurement [23,24]. Due to the cumbersome steps in this method, the algorithm is complex and time-consuming. The measurement methods proposed in the above literature have low accuracy or time-consuming extended defects. A new measurement method with low complexity and high accuracy is urgently needed.

To improve the speed and accuracy of curve segmentation and length measurement, we first convert the “multimodal histogram” into a “quasi-bimodal histogram” to quickly determine the threshold and segment the target curve. Then we refine the curve and accurately calculate the length of the skeleton through the 8-neighborhood features of the pixels on the structure. Finally, we calculate the length of the target curve through the size transformation. The main contributions of this paper are as follows:(1)We propose the QBTS algorithm based on a grayscale histogram, which can quickly and accurately segment the target curve from the neon light design renderings with background interference.(2)We propose the SPSLM algorithm based on the 8-neighborhood model, which improves the accuracy of irregular curve length measurement.(3)We constructed three new image datasets for performance testing of the two proposed algorithms.

The rest of the paper is organized as follows. The steps of the proposed method are described in Section 2. Section 3 discusses experiments on two original datasets in this paper and analyzes the experimental results. Section 4 summarizes the work of this paper.

## 2. Proposed Method

Given a design rendering, we propose a curve segmentation and length measurement method, as shown in Figure 1. The method includes three steps: “Image Preprocessing,” “Threshold Segmentation,” and “Length Measurement.” We first obtain the grayscale image, get the segmentation threshold, and measure the length according to the curve skeleton. The method is described in detail below.

### 2.1. Image Preprocessing

The object of the preprocessing is the original neon design rendering. The goal of the preprocessing is to obtain a grayscale image for threshold segmentation, including the three main steps of color space conversion, channel separation, and grayscale processing.

#### 2.1.1. Image Color Space Conversion

On the one hand, the color space description should conform to the visual perception characteristics of the human eye, and on the other hand, it should be convenient for image processing. The design renderings are usually in RGB color space, but the color space is a non-uniform color space [25]. The color of pixels in this color space are far from the perception of human eyes, so it is not suitable for color image segmentation. However, the HSV color space is a uniform color space that reflects the human visual perception of color. Its V component has nothing to do with the color information of the image, and the H and S components are closely related to the way people perceive color. Therefore, images are converted from RGB color space to HSV color space by linear or non-linear transformations [26]. 

We can convert the (R, G, B) coordinates of a point in RGB color space to (H, S, V) coordinates in HSV color space using the following formula:(1)$$H=\{\begin{array}{c} \arccos\left\{\frac{\left[\left(R\;-\;G\right)+\frac{R\;-\;B}{2}\right]}{{\left[{\left(R\;-\;G\right)}^{2}+\left(R\;-\;B\right)\left(G\;-\;B\right)\right]}^{\frac{1}{2}}}\right\}\;B\leq G\\ 2\pi \;-\;\arccos\left\{\frac{\left[\left(R\;-\;G\right)+\frac{R\;-\;B}{2}\right]}{{\left[{\left(R\;-\;G\right)}^{2}+\left(R\;-\;B\right)\left(G\;-\;B\right)\right]}^{\frac{1}{2}}}\right\}\;B>G \end{array}$$
(2)$$S=\frac{\max\left(R,\;G,\;B\right)-\;\min(R,\;G,\;B)}{\max(R,\;G,\;B)}$$
(3)$$V=\frac{\max\left(R,\;G,\;B\right)}{255}$$

Images before and after the conversion of a design drawing are shown in Figure 2.

#### 2.1.2. HSV Image Channel Separation and Grayscale Processing

Channel separation is the separating of a multi-channel composite image into multiple single-channel photos. Each single-channel image represents a feature of the multi-channel composite image. The HSV image consists of three single-channel images of H, S, and V, representing the three characteristics of the image’s Hue, Saturation, and Value. In the neon design renderings, the target line representing the neon light strip is brighter than the background noise, so we can select the V-channel image, representing the feature of “brightness,” to remove the background noise. The three single-channel images after channel separation are shown in Figure 3.

Although the V-channel image obtained by channel separation can be used as a grayscale image, the image quality is low. To obtain a high-quality grayscale image, the common method is to use RGB as an intermediate and use the following equation to perform grayscale processing.
(4)Gray(x, y)=0.299R(x, y)+0.587G(x, y)+0.114B(x, y)where Gray(x, y) represents the grayscale value of the pixel, whose coordinates are (x, y) on the image after grayscale processing, and R(x, y), G(x, y), and B(x, y) represent the pixel’s R, G, and B channel components. The result of the grayscale processing is shown in Figure 4.

### 2.2. Curve Extraction

Due to the variety of background interference in the neon light design, the grayscale histogram presents multimodal characteristics. Since the brightness of the target curve is higher than the background noise, the peaks of the target curve are always located at the far right of the histogram, and we can regard the remaining peaks as “background peaks.” The entire histogram presents a “quasi-bimodal” feature. 

This paper proposes a QBTS algorithm based on the “quasi-bimodal” feature of the grayscale histogram. The algorithm mainly includes obtaining the grayscale distribution chart and the segmentation threshold. We first obtain the grayscale histogram of the grayscale image and use the sliding smoothing filter to convolve it to get the grayscale distribution map, and then obtain the segmentation threshold by analyzing the characteristics of the peaks and troughs in the grayscale distribution chart. The specific implementation steps are shown in Figure 5.

#### 2.2.1. Get Grayscale Distribution Chart by Sliding Filter Method

Perform a histogram analysis in Figure 4, and obtain its grayscale histogram as shown in Figure 6. The x-axis represents the grayscale value, and the y-axis represents the total number of pixels corresponding to each grayscale value in the grayscale image. Assuming that $N_{\left[1\times 256\right]}$ is the histogram vector of the grayscale image, [1×256] is the size of the histogram vector, and $n_{i}$ represents the number of pixels whose grayscale value is i, then
(5)$$N_{\left[1\times 256\right]}=\left[n_{0}{,\;n}_{1}{,\;n}_{2}{,\;n}_{3}{,\;\ldots \ldots \;n}_{254}{,\;n}_{255}\right]$$

The grayscale histogram has many spikes, so it is filtered by sliding average, and a suitable convolution kernel is selected for linear convolution to make the grayscale histogram smoother. The convolution kernels chosen in this paper are
(6)$$H_{\left[1\times 20\right]}=\left[\frac{1}{20},\;\frac{1}{20},\;\frac{1}{20}\;\ldots \ldots \;\frac{1}{20}\right]$$

The smoothed new histogram vector is
(7)$$N_{\left[1\times 256\right]}^{\prime }=\left[n_{0}^{\prime }{,\;n}_{1}^{\prime }{,\;n}_{2}^{\prime }{,\;n}_{3}^{\prime }{,\;\ldots \ldots \;n}_{254}^{\prime }{,\;n}_{255}^{\prime }\right]$$

The elements $n_{i}^{\prime }$ in the new histogram vector satisfy the following formula:(8)$$n_{i}^{\prime }=\{\begin{array}{ll} \overset{19}{\underset{k=0}{\sum }}n_{i+k}\cdot H_{\left[1\times 20\right]}\left[k\right], & 0\leq i\;\leq 236\\ \overset{255-i}{\underset{k=0}{\sum }}n_{i+k}\cdot H_{\left[1\times 20\right]}\left[k\right]+\overset{i+237}{\underset{k=0}{\sum }}n_{k}\cdot H_{\left[1\times 20\right]}\left[255-i+k\right] & \;237\leq i\leq 255 \end{array}\;$$

The grayscale distribution after the smoothing process is shown in Figure 7.

#### 2.2.2. Get the Segmentation Threshold by the Quasi-Bimodal Characteristics of the Gray Distribution Chart

After obtaining the grayscale distribution of the V (Value) channel image, the segmentation threshold is obtained according to the basic idea of the bimodal method. First, mark the “target peak” and the “background peak” in the grayscale distribution chart, and then select the trough between the two peaks as the threshold. Since the “value” of the target curve is the highest, the rightmost peak in the grayscale distribution graph is marked as the “target peak.” Then the mountain with the most prominent peak among the remaining peaks is chosen as the “background peak.” When there is more than one trough between the “target peak” and the “background peak,” the trough that is closest to the “target peak” and whose amplitude is less than the average of all the troughs is marked as the threshold value. 

According to the above steps, mark the “target peak,” “background peak,” and “threshold value” in the grayscale distribution diagram, as shown in Figure 8. As can be seen from Figure 8, the “target peak,” “background peak,” and “threshold” are “245”, “97,” and “208,” respectively.

We segment Figure 4 with the threshold marked in Figure 8 and get the target curve, as shown in Figure 9.

### 2.3. Length Measurement

#### 2.3.1. Curve Refinement

The image refinement of the binary image obtained by threshold segmentation can obtain the single-pixel skeleton of the target curve. We use the improved Zhang-Suen refinement algorithm to refine the target curve [27]. On the one hand, the algorithm has simple logic and a fast running speed. On the other hand, it also overcomes the disadvantage of missing some pixels in the traditional Zhang-Suen algorithm, resulting in partially refined textures that are not single pixels.

Using this algorithm to refine Figure 9, we can obtain a single-pixel skeleton image, as shown in Figure 10.

#### 2.3.2. Skeleton Length Measurement

After obtaining the single-pixel skeleton of the target curve, the method of directly counting the number of pixels as the skeleton length has a large error [15,16,17]. To improve the measurement accuracy, this paper proposes a single-pixel skeleton length measurement (SPSLM) algorithm based on the 8-neighborhood model. We can label the 8 pixels adjacent to the pixel $p_{1}$ and use the schematic diagram shown in Figure 11 to represent the 8-neighborhood model of $p_{1}$. The specific implementation steps are shown in Figure 12.

${B(p}_{1})$, N, and C are three parameters determined by the 8-neighborhood of $p_{1}$, as shown in Figure 12. ${B(p}_{1})$ represents the number of foreground pixels in the 8-neighborhood of $p_{1}$. N and C represent the number of foreground pixels among the 4 pixels directly adjacent and diagonally adjacent to $p_{1}$, respectively.

According to the 8-neighborhood model of $p_{1}$ shown in Figure 11, ${B(p}_{1})$, N, and C can be represented by the following equations.
(9)$$\{\begin{array}{c} B\left(p_{1}\right){=p}_{2}{+p}_{3}{+p}_{4}{+p}_{5}{+p}_{6}{+p}_{7}{+p}_{8}{+p}_{9}\\ {N=p}_{2}{+p}_{4}{+p}_{6}{+p}_{8}\\ {C=p}_{3}{+p}_{5}{+p}_{7}{+p}_{9} \end{array}$$

Next, we discuss the connection between $L_{p_{1}}$ and the three parameters of ${B(p}_{1})$, N, and C. It is important to note that the following discussion of the distribution of pixels in the 8-neighborhood of pixel $p_{1}$ has removed the situation that meets the labeling conditions of the improved Zhang-Suen refinement algorithm. Among them, A denotes the length of a single pixel, B denotes the diagonal measurement of a single pixel, and $L_{p_{1}}$ denotes the actual length represented by the pixel $p_{1}$.

According to the different values of ${B(p}_{1})$, the 8-neighborhood model of $p_{1}$ is classified and discussed. The model diagrams are shown in Appendix A.

If $B\left(p_{1}\right)=1$, as shown in Figure A1, then there are



(10)
$$\{\begin{array}{c} N=1,\;C=0:L_{p_{1}}=1\;*\;A+0\;*\;B\\ N=0,\;C=1:L_{p_{1}}=0\;*\;A+1\;*\;B \end{array}\;\Rightarrow {\;L}_{p_{1}}=N\;*\;A+C\;*\;B$$



If $B\left(p_{1}\right)=2$, as shown in Figure A2, then there are 



(11)
$$\{\begin{array}{c} N=2,\;C=0:L_{p_{1}}=2\;*\;\frac{A}{2}+0\;*\;\frac{B}{2}\\ N=1,\;C=1:L_{p_{1}}=1\;*\;\frac{A}{2}+1\;*\;\frac{B}{2}\\ N=0,\;C=2:L_{p_{1}}=0\;*\;\frac{A}{2}+2\;*\;\frac{B}{2} \end{array}\;\Rightarrow \;L_{p_{1}}=N\;*\;\frac{A}{2}+C\;*\;\frac{B}{2}$$



If $B\left(p_{1}\right)=3$, as shown in Figure A3, then there are 



(12)
$$\{\begin{array}{c} N=3,\;C=0:L_{p_{1}}=2\;*\;\frac{A}{2}+0\;*\;\frac{B}{2}\\ N=2,\;C=1:L_{p_{1}}=2\;*\;\frac{A}{2}+0\;*\;\frac{B}{2}\\ N=1,\;C=2:L_{p_{1}}=1\;*\;\frac{A}{2}+1\;*\;\frac{B}{2}\\ N=0,\;C=3:L_{p_{1}}=0\;*\;\frac{A}{2}+2\;*\;\frac{B}{2} \end{array}\;\Rightarrow \;\{\begin{array}{c} N\;>\;C:L_{p_{1}}=2\;*\;\frac{A}{2}+0\;*\;\frac{B}{2}\\ 0\;<\;N\;<\;C:L_{p_{1}}=1\;*\;\frac{A}{2}+1\;*\;\frac{B}{2}\\ N\;=\;0:L_{p_{1}}=0\;*\;\frac{A}{2}+2\;*\;\frac{B}{2} \end{array}$$



If $B\left(p_{1}\right)=4$, as shown in Figure A4, then there are




(13)
$$\{\begin{array}{c} N=4,\;C=0:L_{p_{1}}=2\;*\;\frac{A}{2}+0\;*\;\frac{B}{2}\\ N=3,\;C=1:L_{p_{1}}=2\;*\;\frac{A}{2}+0\;*\;\frac{B}{2}\\ N=2,\;C=2:L_{p_{1}}=1\;*\;\frac{A}{2}+1\;*\;\frac{B}{2}\\ N=1,\;C=3:L_{p_{1}}=0\;*\;\frac{A}{2}+2\;*\;\frac{B}{2}\\ N=0,\;C=4:L_{p_{1}}=0\;*\;\frac{A}{2}+2\;*\;\frac{B}{2} \end{array}\;\Rightarrow \;\{\begin{array}{c} N\;>\;C:L_{p_{1}}=2\;*\;\frac{A}{2}+0\;*\;\frac{B}{2}\\ N\;=\;C:L_{p_{1}}=1\;*\;\frac{A}{2}+1\;*\;\frac{B}{2}\\ N\;<\;0:L_{p_{1}}=0\;*\;\frac{A}{2}+2\;*\;\frac{B}{2} \end{array}$$



If $B\left(p_{1}\right)=5$, as shown in Figure A5, then there are




(14)
$$\{\begin{array}{c} N=4,\;C=1:L_{p_{1}}=2\;*\;\frac{A}{2}+0\;*\;\frac{B}{2}\\ N=3,\;C=2:L_{p_{1}}=2\;*\;\frac{A}{2}+0\;*\;\frac{B}{2}\\ N=2,\;C=3:L_{p_{1}}=0\;*\;\frac{A}{2}+2\;*\;\frac{B}{2}\\ N=1,\;C=4:L_{p_{1}}=0\;*\;\frac{A}{2}+2\;*\;\frac{B}{2} \end{array}\;\Rightarrow \;\{\begin{array}{c} N\;>\;C:L_{p_{1}}=2\;*\;\frac{A}{2}+0\;*\;\frac{B}{2}\\ N\;<\;C:L_{p_{1}}=0\;*\;\frac{A}{2}+2\;*\;\frac{B}{2} \end{array}$$



If $B\left(p_{1}\right)=6$, as shown in Figure A6, then there are




(15)
$$\{\begin{array}{c} N=4,\;C=2:L_{p_{1}}=2\;*\;\frac{A}{2}+0\;*\;\frac{B}{2}\\ N=3,\;C=3:L_{p_{1}}=1\;*\;\frac{A}{2}+1\;*\;\frac{B}{2}\\ N=2,\;C=4:L_{p_{1}}=0\;*\;\frac{A}{2}+2\;*\;\frac{B}{2} \end{array}\;\Rightarrow \;\{\begin{array}{c} N\;>\;C:L_{p_{1}}=2\;*\;\frac{A}{2}+0\;*\;\frac{B}{2}\\ N\;=\;C:L_{p_{1}}=1\;*\;\frac{A}{2}+1\;*\;\frac{B}{2}\\ N\;<\;0:L_{p_{1}}=0\;*\;\frac{A}{2}+2\;*\;\frac{B}{2} \end{array}$$



If $B\left(p_{1}\right)=7$, as shown in Figure A7, then there are 



(16)
$$\{\begin{array}{c} N=4,\;C=3:L_{p_{1}}=2\;*\;\frac{A}{2}+0\;*\;\frac{B}{2}\\ N=3,\;C=4:L_{p_{1}}=0\;*\;\frac{A}{2}+2\;*\;\frac{B}{2} \end{array}\;\Rightarrow \;\{\begin{array}{c} N\;>\;C:L_{p_{1}}=2\;*\;\frac{A}{2}+0\;*\;\frac{B}{2}\\ N\;<\;C:L_{p_{1}}=0\;*\;\frac{A}{2}+2\;*\;\frac{B}{2} \end{array}$$



If $B\left(p_{1}\right)=8$, as shown in Figure A8, then there are



(17)
$$N=4,\;C=4:L_{p_{1}}=0\;*\;\frac{A}{2}+2\;*\;\frac{B}{2}\;\Rightarrow {\;L}_{p_{1}}=0\;*\;\frac{A}{2}+2\;*\;\frac{B}{2}$$



Traverse the single-pixel skeleton image and assume that the number of pixels that satisfy the condition of $B\left(p_{1}\right)=i$ is $N_{i}$ and the pixel length of the target curve skeleton is $L_{\mathrm{TP}}$, then there is the following formula:(18)$$L_{\mathrm{TP}}=\overset{8}{\underset{i=1}{\sum }}\overset{N_{i}}{\underset{1}{\sum }}L_{p_{1}}$$

#### 2.3.3. Size Transformation

Assumption α represents the scale factor of the pixel size to the actual size in cm/pixel. Set the length of a single-pixel A to “unit1”, then the proper length of the pixel skeleton can be calculated by the following formula.
(19)$$L_{\mathrm{TR}}{=\alpha \;\times \;L}_{\mathrm{TP}}=\alpha \;\times \;\overset{8}{\underset{i=1}{\sum }}\overset{N_{i}}{\underset{1}{\sum }}L_{p_{1}}$$

## 3. Experiments and Results

This section shows the experimental results of the proposed QBTS algorithm and SPSLM algorithm on the original datasets and compares them with the results of other algorithms. Furthermore, all experiments were performed on an Intel Core i5-9400 2.9 GHz desktop with 8 GB of RAM.

### 3.1. Performance Metrics

To evaluate the proposed method, we choose accuracy and running speed as evaluation metrics. The segmentation accuracy of the QBTS algorithm is defined as ${\mathrm{Acc}}_{S}=\frac{N_{\mathrm{same}}}{N_{\mathrm{total}}}$, where $N_{\mathrm{same}}$ represents the number of pixels with the same pixel value in the binary image obtained after image segmentation by the QBTS algorithm and the binary image of the standard segmented image, and $N_{\mathrm{total}}$ represents the total number of pixels. According to the expression of ${\mathrm{Acc}}_{S}$, its range is [0, 1]. The measurement accuracy of the SPSLM algorithm is defined as ${\mathrm{Acc}}_{M}=\frac{L_{M}}{L_{R}}$, where $L_{M}$ represents the length measurement value of the SPSLM algorithm, and $L_{R}$ represents the reference value obtained by manual measurement. Since $L_{R}$ is a manual measurement value, there is also a particular error, so in some cases, the value of ${\mathrm{Acc}}_{M}$ may be greater than 1. For the entire dataset, the closer ${\mathrm{Acc}}_{M}$ is to 1, the higher the measurement accuracy of the SPSLM algorithm.

In addition, the running speed of an algorithm is usually measured in terms of running time. The shorter the time it takes for the algorithm to complete the segmentation or length measurement, the faster it is.

### 3.2. Dataset

We conduct experiments on three original datasets to analyze these two algorithms’ accuracy and running speed. Additionally, all images are less than 2000 × 2000 in size and have different pixel dimensions.

Mini. This dataset contains 12 original neon design renderings, as shown in Figure 13. In addition, the dataset also includes 12 standard target curves that were segmented manually by multiple researchers. This dataset is used to test the segmentation accuracy of the QBTS algorithm.

Neon Rendering. This dataset contains 198 images, all sourced from the Internet. These images are actual neon design renderings with patterned curves and backgrounds with many types of noise inside. This dataset is used to test the running speed of the QBTS algorithm.

Neon Curve. This dataset contains 139 images of neon pattern curves without noise interference and 139 corresponding single-pixel skeleton images. Hunan Kangxuan Technology Co., Ltd. provides the original images and the corresponding curve length value. The single-pixel skeleton image is obtained by refining the original picture through the improved Zhang-Suen algorithm. This dataset is used to test the measurement accuracy and running speed of the SPSLM algorithm.

### 3.3. Experimental Results

#### 3.3.1. Performance Analysis of the QBTS Algorithm

Accuracy of Segmentation

To test the segmentation accuracy of the QBTS algorithm, we compared it with the OTSU algorithm and the bimodal method. We used three threshold segmentation algorithms to segment the images of the mini dataset, and the experimental results are shown in Figure 14. Each image segmentation result consists of four images. From left to right are the original image, the binary image of the standard target curve, and the binary image obtained by dividing the QBTS algorithm, the OTSU algorithm, and the bimodal method, respectively. It can be seen intuitively from Figure 14 that the similarity between the binary image segmented by the QBTS algorithm and the standard binary image is the highest. The images segmented by the other two algorithms still have varying degrees of noise interference, and even the target curve cannot be seen in some segmented images. The results show that the segmentation result of the QBTS algorithm is better, and it can accurately remove noise interference and segment the target curve.

To conduct a more accurate quantitative analysis of the segmentation accuracy of the QBTS algorithm, we calculated the segmentation accuracy of the three algorithms according to the definition of ${\mathrm{Acc}}_{S}$, as shown in Figure 15. The ordinate in the figure is the segmentation accuracy of the three algorithms.

The results show that the average segmentation accuracy of the QBTS algorithm is 97.9%, much higher than the 89.6% and 64.4% of the other two algorithms. At the same time, the distribution of the segmentation accuracy of the algorithm is also more concentrated, indicating that the QBTS algorithm has higher robustness. At the same time, we also noticed that the segmented images obtained using the QBTS algorithm are not entirely accurate. There are two main reasons for this: on the one hand, the QBTS algorithm belongs to the global threshold segmentation algorithm and can’t segment all the boundaries of the target curve very finely and accurately. On the other hand, researchers manually segment the reference images of the dataset, and in this process, errors will inevitably occur and affect the experimental results.

Running Speed of QBTS Algorithm

In addition to testing segmentation accuracy, this paper also conducts experiments on the running speed of the QBTS algorithm on the Neon rendering dataset. Figure 16 shows the time required for each of the three algorithms to segment images in the dataset Neon rendering. To compare the segmentation speed more intuitively, we calculated the ratio of the time required by the QBTS algorithm and the OTSU algorithm for segmentation to the time needed for the Bimodal algorithm, as shown in Figure 17.

We can see from Figure 16 that the time consumed by the QBTS algorithm to segment images is generally shorter than the other two algorithms. In addition, the figure has some discrete points outside the 1.5IQR range. The main reason is that the time complexity of the QBTS algorithm is O(mn), where m×n represents the size of the image. The running time is closely related to the image size, so some images with larger sizes will take longer to process. We can see from Figure 17 that the average values of the split time ratios of the OTSU algorithm, the QBTS algorithm, and the Bimodal algorithm are 0.98 and 0.47, respectively. It shows that for the same image, the segmentation time of the QBTS algorithm is shorter, the segmentation speed is faster, and the segmentation speed has increased by about 50%.

To compare the performance of these three image segmentation algorithms more intuitively and clearly, we summarize and extract the key data from Figure 15, Figure 16 and Figure 17, as shown in Table 1. It can be seen from Table 1 that the performance of the QBTS algorithm is much better than that of the OTSU algorithm and the Bimodal algorithm in terms of average segmentation accuracy and segmentation speed.

According to the above experimental results, the QBTS algorithm proposed in this paper performs better when segmenting the target curve from the neon sign design drawing with complex background noise interference. Compared with the OTSU and Bimodal algorithms, it has better segmentation accuracy, robustness, and faster segmentation speed.

#### 3.3.2. Performance Analysis of the SPSLM Algorithm

Accuracy of Measurement

Figure 18 shows the length measurement accuracy of the SPSLM algorithm and the other four methods. The SPSLM algorithm in the figure is a single-pixel skeleton length measurement algorithm based on the 8-neighborhood model proposed in this paper. Method 1, Method 2, Method 3, and Method 4 are methods for measuring curve lengths used in [13,14,15], [16,17,18,19,20], [12], and [21,22], respectively.

We can see from Figure 18 that the average measurement accuracy of the SPSLM algorithm proposed in this paper is 99.1%, and the average measurement accuracy of the other four methods is 88.3%, 92.5%, 19.9%, and 88.1%, respectively; it shows that the accuracy of the SPSLM algorithm is higher. At the same time, we can also see that some values are greater than 100%. This phenomenon is also consistent with our analysis of ${\mathrm{Acc}}_{M}$.

Running Speed of SPSLM Algorithm

After the accuracy test, we ran the SPSLM algorithm’s speed test on the Neon Curve dataset. Figure 19 shows the running speed of the length measurement algorithms. The results show that the average measurement time of Method 3 is 0.001s, which is much shorter than other methods. Because Method 3 mainly estimates the length through the area range, it has the advantage of low time complexity, but the measurement accuracy is also very low. The average measurement time of Method 4 is 6.90s, much higher than the other four methods. The method includes canny edge detection, image refinement, and skeleton length measurement. The processing process is cumbersome and time-consuming.

Figure 20 shows the running speeds of SPSLM, Method 1, and Method 2 separately. Their average measurement times were 0.615s, 0.565s, and 0.577s, respectively, which were the same and had similar distributions. Because the measurement principle of these three methods is first to perform image refinement to obtain a single-pixel skeleton and then measure the length of the structure, their time complexity is the same, so the measurement speed is the same.

To compare the performance of these five length measurement methods more intuitively and clearly, we summarize and extract the pivotal data from Figure 18, Figure 19 and Figure 20, as shown in Table 2. As can be seen from Table 2, in terms of measurement accuracy, the average accuracy of the SPSLM algorithm is much higher than that of other algorithms. In terms of measurement speed, on the premise of ensuring the necessary accuracy, the average running speed of the SPSLM algorithm is comparable to Method 1 and Method 2, and both are much faster than Method 4.

Based on the above analysis of the running speed and the measurement accuracy, the SPSLM algorithm dramatically improves the measurement accuracy while maintaining a low algorithm complexity, and its overall performance is better than the other length measurement methods.

## 4. Conclusions

This paper used digital image processing techniques to measure the length of irregular curves in neon design renderings. Firstly, a new QBTS algorithm was proposed to segment and extract the target curve. Then, a single-pixel skeleton length measurement algorithm based on the 8-neighborhood model was proposed to measure the length of the skeleton of the target curve. Finally, we conducted tests on the three original datasets of this paper, respectively. The results showed that the average segmentation accuracy of the QBTS algorithm was 97.9%, and the segmentation speed was more than 50% higher than the other two algorithms. The average measurement accuracy of the length measurement algorithm was 99.1%, higher than the four existing length measurement algorithms, and the measurement speed was comparable.

The above results demonstrate that the two algorithms proposed in this paper can be used for the problem of “accurately segmenting irregular target curves from images with background interference and measuring their length accurately”, and can be applied in engineering practice. Subsequent research will further improve the segmentation accuracy and applicability of the threshold segmentation algorithm, laying a solid foundation for future applications in more areas of target curve segmentation.

## Figures and Tables

**Figure 1 sensors-22-05761-f001:**
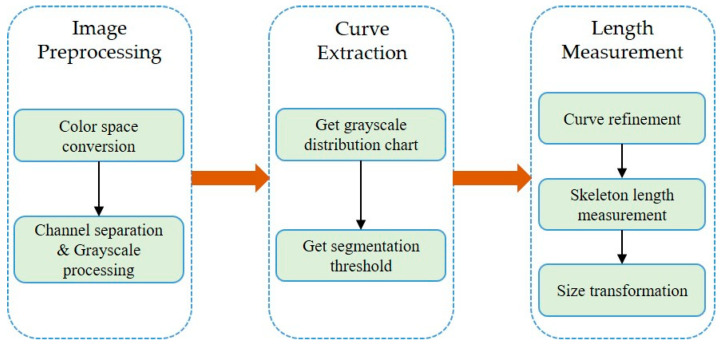
Steps of the proposed method.

**Figure 2 sensors-22-05761-f002:**
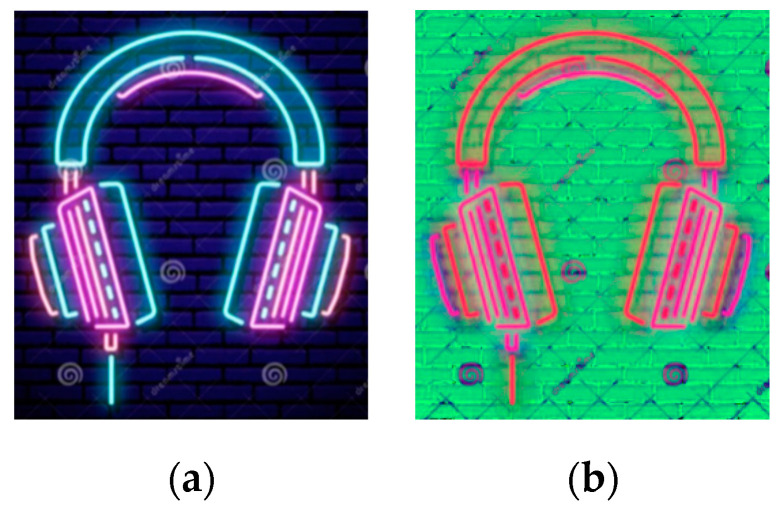
A rendering of a neon design before and after color space conversion: (**a**) The design rendering under the RGB color model; (**b**) The design rendering under the HSV color model.

**Figure 3 sensors-22-05761-f003:**
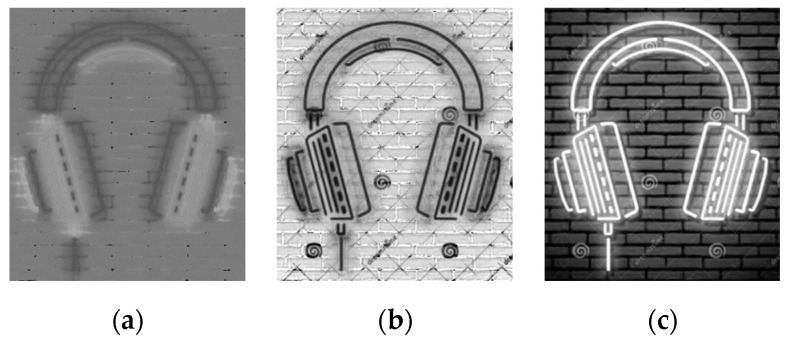
Three single-channel images: (**a**) The H channel image; (**b**) The S channel image; (**c**) The V channel image.

**Figure 4 sensors-22-05761-f004:**
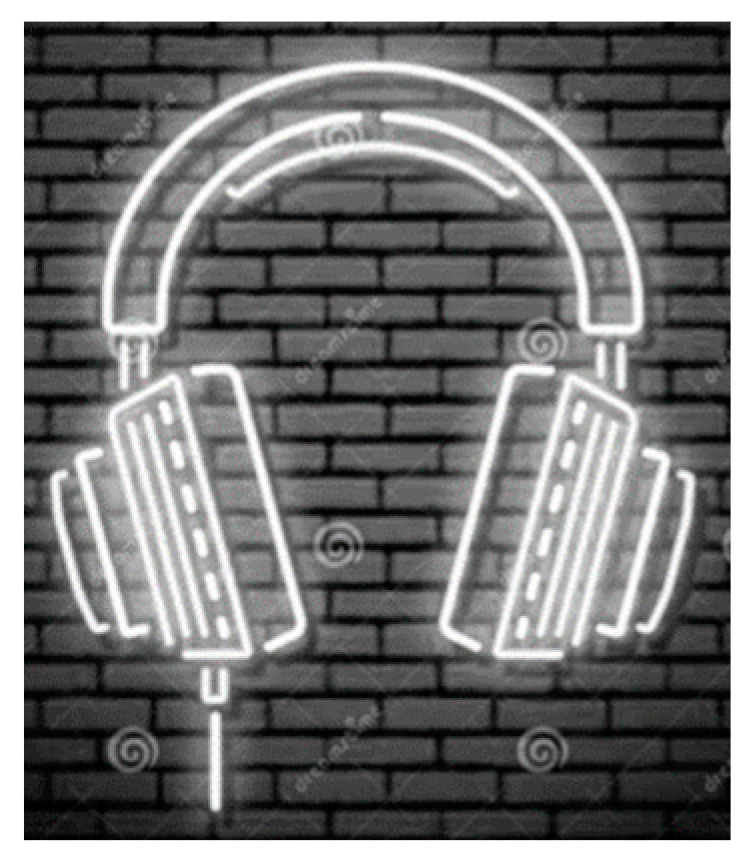
Grayscale image of the V-channel.

**Figure 5 sensors-22-05761-f005:**
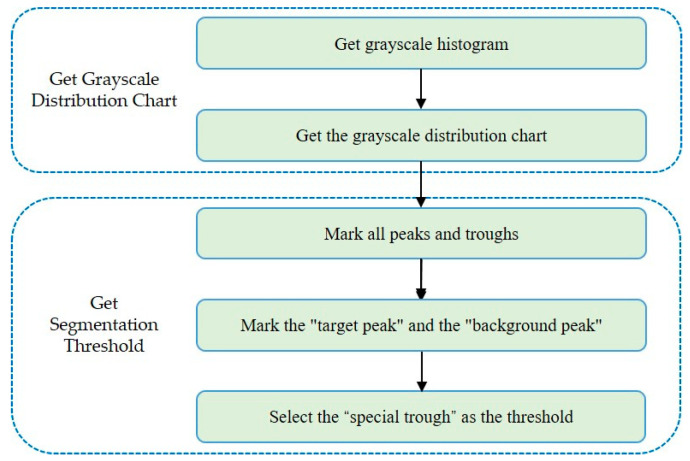
Implementation steps of the QBTS algorithm.

**Figure 6 sensors-22-05761-f006:**
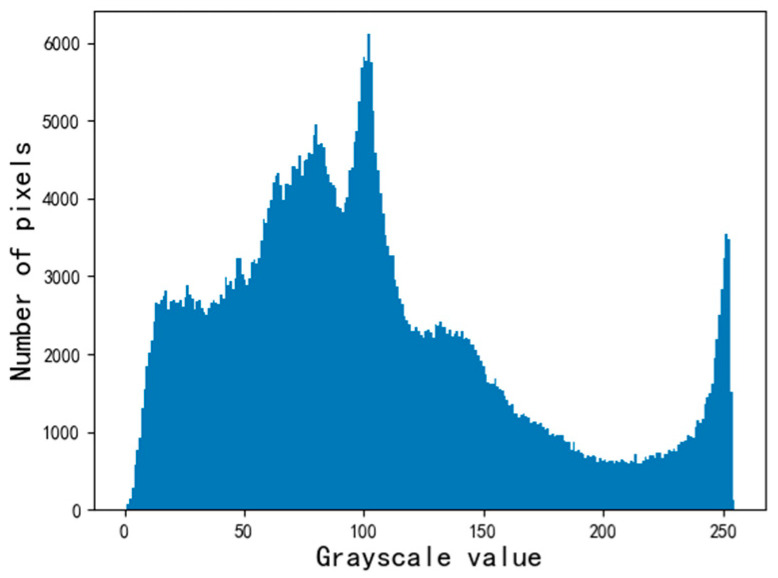
Histogram of grayscale.

**Figure 7 sensors-22-05761-f007:**
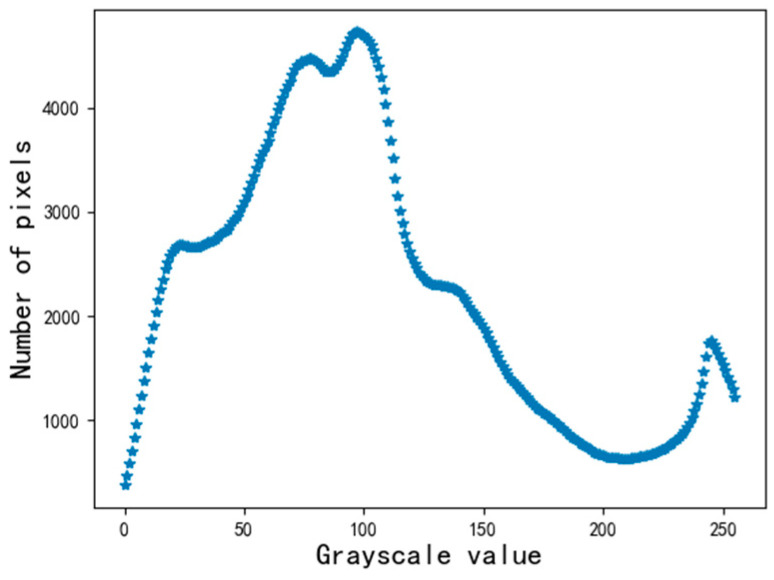
Distribution chart of grayscale.

**Figure 8 sensors-22-05761-f008:**
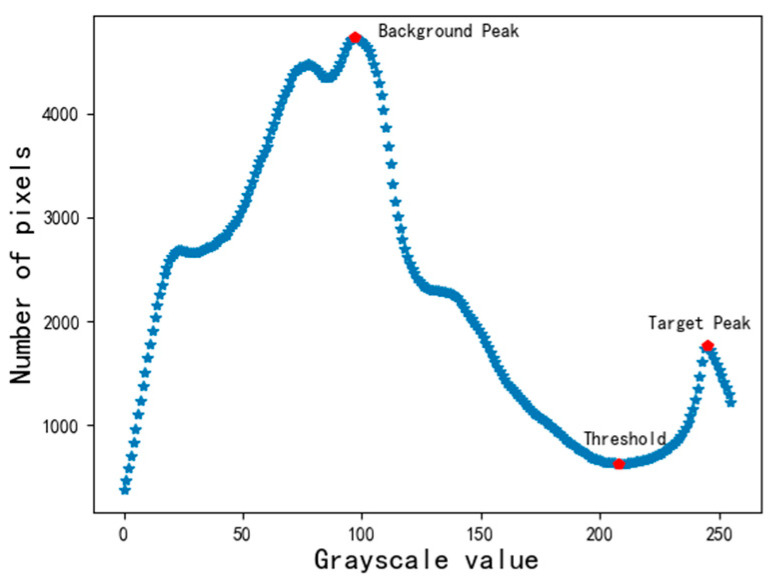
Location of “Background Peak,” “Target Peak,” and “Threshold” in the grayscale distribution chart.

**Figure 9 sensors-22-05761-f009:**
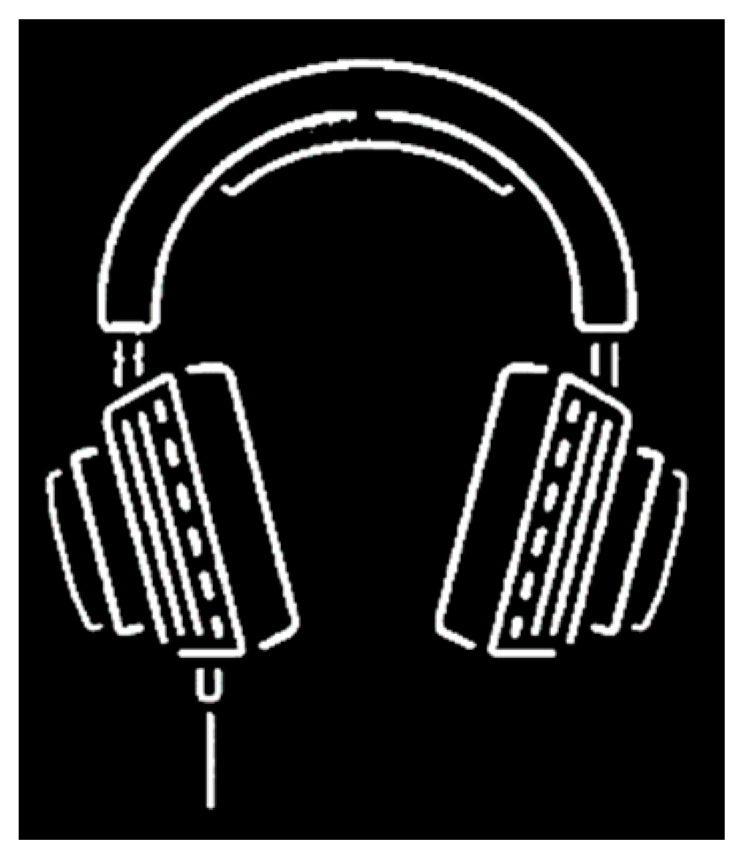
Binary image segmented by QBTS algorithm.

**Figure 10 sensors-22-05761-f010:**
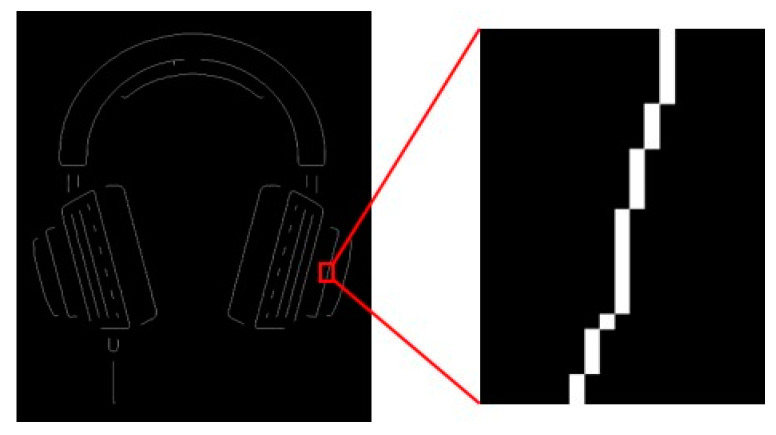
The single-pixel skeleton of the target curve.

**Figure 11 sensors-22-05761-f011:**
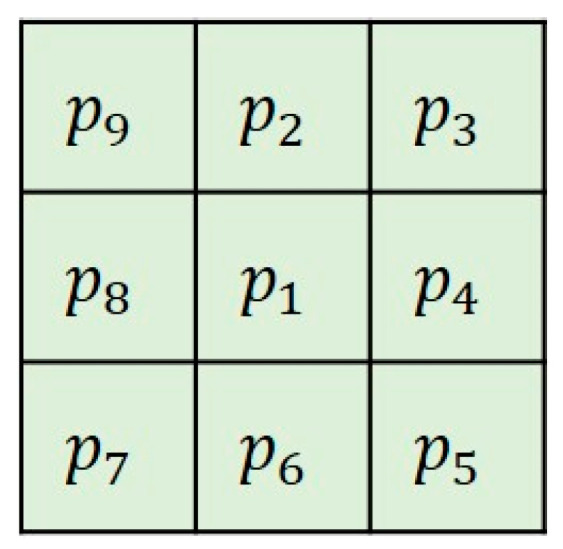
The 8-neighborhood model of pixel $p_{1}$.

**Figure 12 sensors-22-05761-f012:**
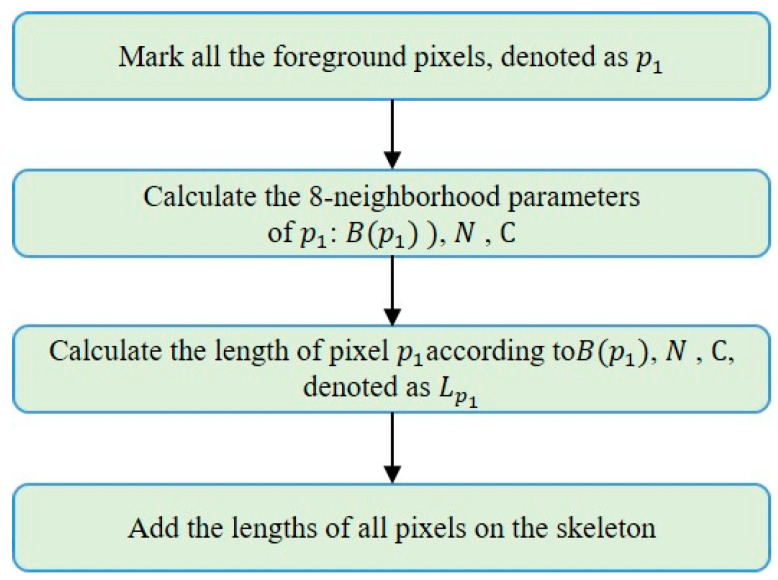
Steps of single-pixel skeleton length measurement algorithm.

**Figure 13 sensors-22-05761-f013:**
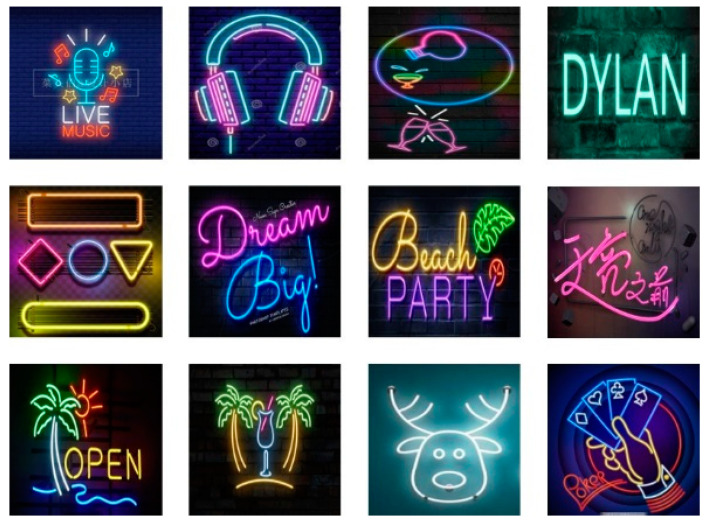
Neon design renderings included in the mini dataset.

**Figure 14 sensors-22-05761-f014:**
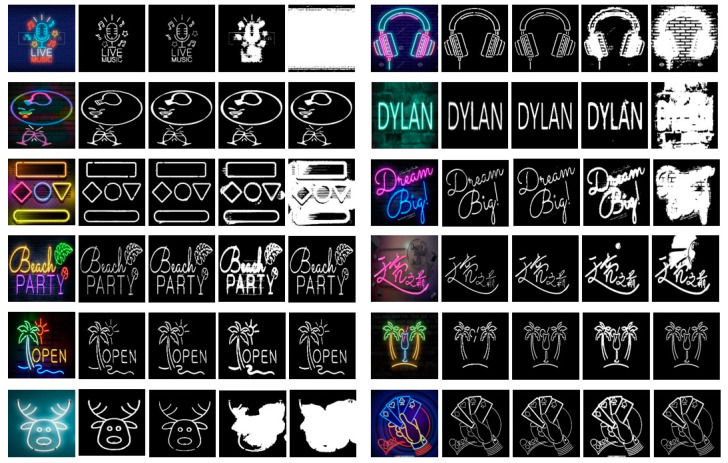
The original images, the standard binary images and the binary images are obtained by dividing by the QBTS algorithm, the OTSU algorithm, and the bimodal method, respectively.

**Figure 15 sensors-22-05761-f015:**
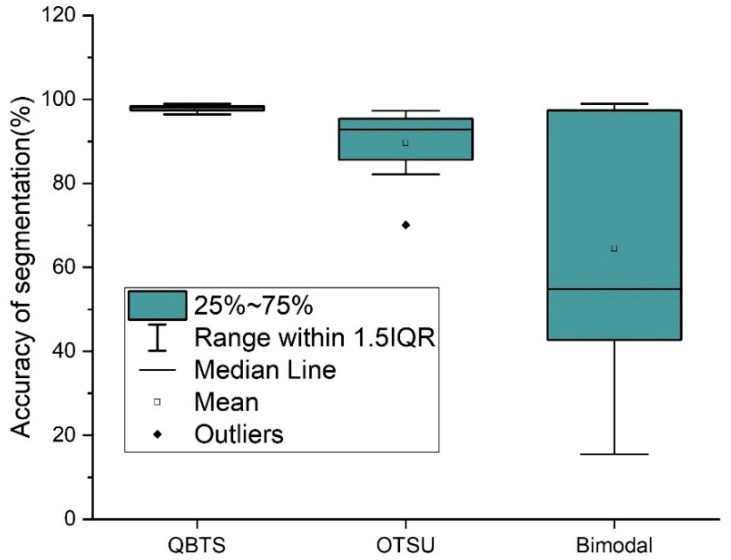
Comparison of accuracy of three threshold segmentation algorithms.

**Figure 16 sensors-22-05761-f016:**
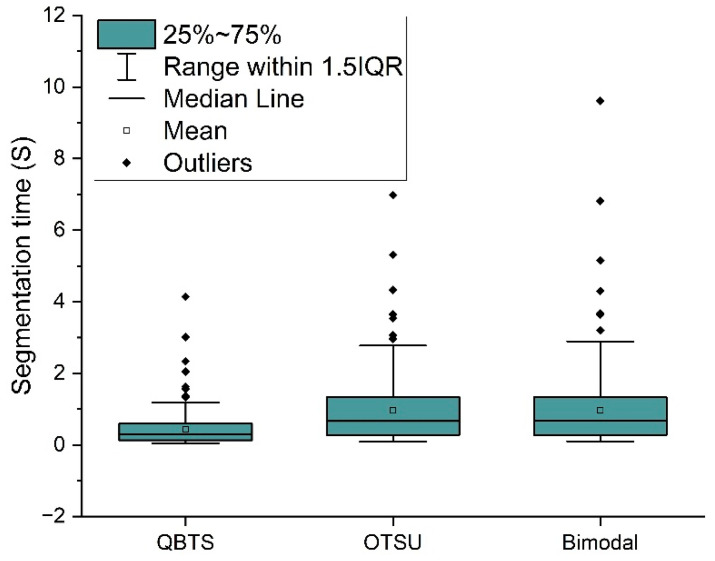
Comparison of running time of three threshold segmentation algorithms.

**Figure 17 sensors-22-05761-f017:**
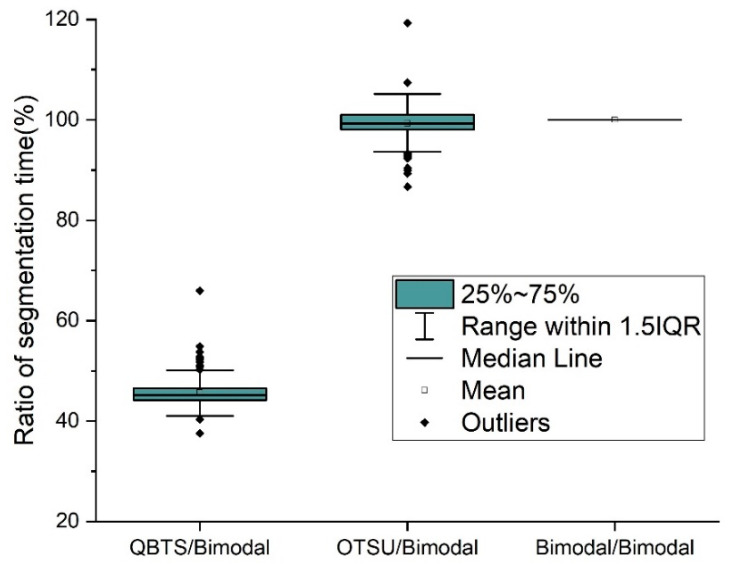
The ratio of running time for the QBTS algorithm, OTSU algorithm, and bimodal algorithm.

**Figure 18 sensors-22-05761-f018:**
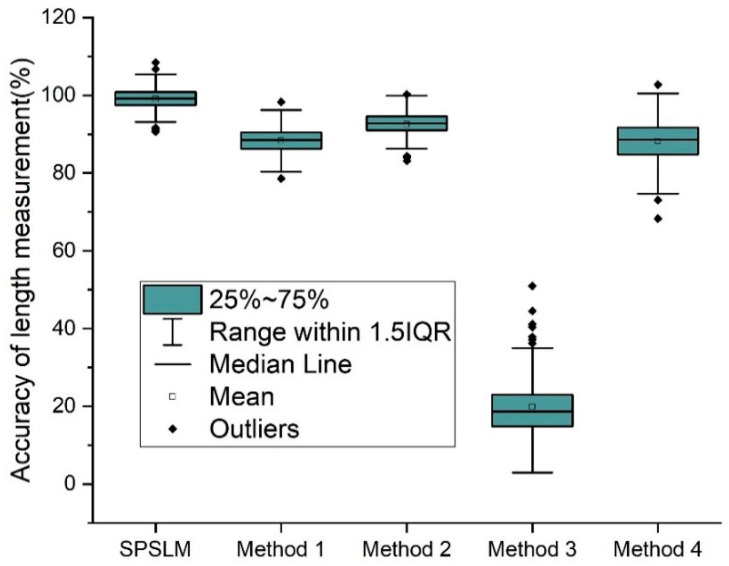
Accuracy of different length measurement algorithms.

**Figure 19 sensors-22-05761-f019:**
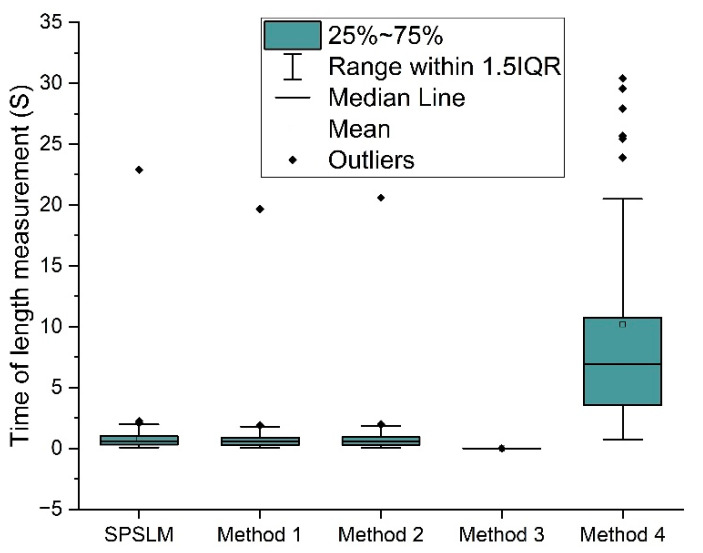
Running time of different length measurement methods.

**Figure 20 sensors-22-05761-f020:**
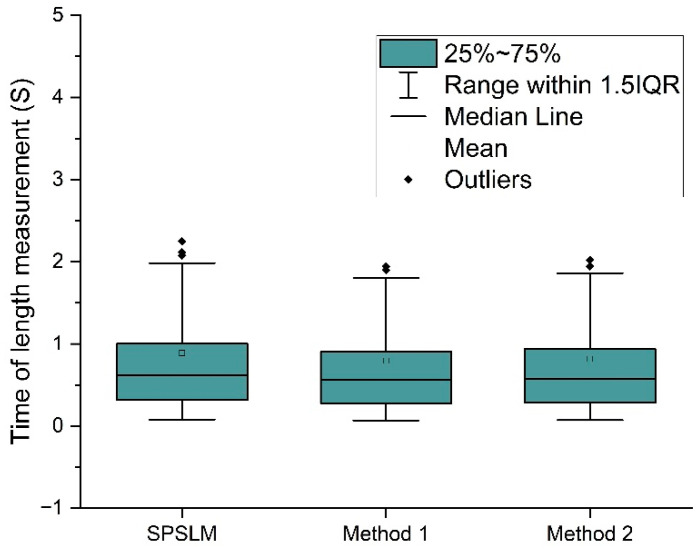
Running time of SPSLM, Method 1 and Method 2.

**Table 1 sensors-22-05761-t001:** Summary and comparison of average accuracy and running speed of three threshold segmentation algorithms.

Method	Average Accuracy (%)	Time / Time Bimodal
QBTS	97.9	0.47
OTSU	89.6	0.98
Bimodal	64.4	1

**Table 2 sensors-22-05761-t002:** Summary and comparison of average accuracy and running time of five length measurement algorithms.

Method	Average Accuracy (%)	Average Running Time (S)
SPSLM	99.1	0.615
Method 1	88.3	0.565
Method 2	92.5	0.577
Method 3	19.9	0.001
Method 4	88.1	6.90

## Data Availability

The datasets that support the findings of this study are openly available at [https://github.com/csuruan/Image-Processing-Datasets-Ruan.git, accessed on 28 June 2022]. The datasets used in this paper are all original datasets by the author. Some of the pictures included in the data set are from public resources on the Internet, and the rest are from Hunan Kangxuan Technology Co., Ltd. All images obtained from public sources on the Internet are for scientific research purposes only and do not involve commercial interests.

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
