# Peer review of "Threshold Segmentation and Length Measurement Algorithms for Irregular Curves in Complex Backgrounds"

_sensors, 2022, doi:10.3390/s22155761_

Round 1
Reviewer 1 Report
line 140 eq 4 reference to coefficients?
line162.title is not a method
ine179.title is not a method
Author Response
Dear Reviewer:
Thank you very much for your review of my article and your valuable suggestions. After carefully reading your suggestions for revision, I have revised and improved the article. Now I will reply to your questions and suggestions.
Point 1: line 140 eq 4 reference to coefficients?
Response 1: Generally speaking, there are two kinds of coefficients of the weighted average grayscale method, as shown in equation (a) and equation (b). The former is the grayscale weight used by the OpenCV open library, and the latter is a grayscale weight proposed from the perspective of human physiology. The weight coefficient of formula (b) is more in line with the visual characteristics of the human eye and is convenient for subsequent image segmentation. So the equation 4 on line 150 of the revised paper adopts the second weight.
Point 2: Line162.title is not a method.
Response 2: According to your suggestion, I summarized the method to get the grayscale distribution chart and modified the title to "Get Grayscale Distribution Chart by Sliding Filter Method." The new title is on line 172 of the revised article.
Point 3: Line179.title is not a method.
Response 3: According to your suggestion, I summarized the method to get the grayscale distribution chart and modified the title to " Get the Segmentation Threshold by the Quasi-bimodal Characteristics of the Gray Distribution Chart." The new title is on line 190 of the revised article.
Thanks again for taking the time to review my article.
Yours sincerely,
Xusheng Ruan
Reviewer 2 Report
Some comments as follow:
1. I only see that the author uses the new method to get the accuracy of the graphs of the datasets, how do you get the exact lengths of the images in these datasets? Do you compare the full length of the figure or just a small section?
2. Why use HSV images for processing? Why not directly use RGB images to convert to grayscale images for processing?
3. If the actual size of the pixels of the image is not constant, is this method still be applied?
4. The experimental comparison needs to add some tables to summarize and compare with other different methods.
Author Response
Dear Reviewer:
Thank you very much for your review of my article and your valuable suggestions. After carefully reading your suggestions for revision, I have revised and improved the article. Now I will reply to your questions and suggestions.
Point 1: I only see that the author uses the new method to get the accuracy of the graphs of the datasets, how do you get the exact lengths of the images in these datasets? Do you compare the full length of the figure or just a small section?
Response 1: In lines 299 – 301 of the revised article, we mentioned that the dataset Neon Curve comes from Hunan Kangxuan Technology Co., Ltd. All images in this dataset are neon design pattern images processed during the company's actual production process. The length of the curve on the image is generally measured manually by several staff members to obtain the average value, and then the measured value is corrected according to the length of the neon light strip consumed in actual production, and finally the accurate value of the curve length is obtained. So we compare the full length of the figure, not just a small section.
Point 2: Why use HSV images for processing? Why not directly use RGB images to convert to grayscale images for processing?
Response 2: Thank you so much for asking this question and making me rethink and refine it. I have explained this issue in the manuscript of an older article, but the formulation is less clear. I added new explanations to lines 121 to 130 of the revised paper.
On the one hand, the color space description should conform to the visual perception characteristics of the human eye, and on the other hand, it should be convenient for image processing. The RGB color space is a non-uniform color space. The color of pixels in this color space is far from the perception of human eyes, so it is not suitable for color image segmentation. However, the HSV color space is a uniform color space that reflects the human visual perception of color. Its V component has nothing to do with the color information of the image, and the H and S components are closely related to the way people perceive color. Therefore, this paper chooses to convert the image to the HSV color space for segmentation processing, rather than directly segmenting it in the RGB space.
Point 3: If the actual size of the pixels of the image is not constant, is this method still be applied?
Response 3: This method can still be applied.
The method proposed in this paper consists of two main steps: image segmentation and curve length measurement. When segmenting an image, in order to more accurately grasp the overall characteristics of the image, the QBTS algorithm scans all the pixels on the grayscale image by traversing to obtain the grayscale histogram of the image and finally determines the segmentation threshold. When measuring the length of a single-pixel skeleton, the SPSLM algorithm needs to traverse all the pixels on the skeleton, analyze the 8-neighborhood features, and finally calculate the skeleton length. Both algorithms involve traversal operations and are not affected by the pixel dimensions of the image. In addition, as mentioned on lines 288 and 289 of the revised paper, the image pixel size of the dataset is not constant, but the method still has good experimental results. So, this method can still be applied even if the actual size of the image pixels is not constant.
Point 4: The experimental comparison needs to add some tables to summarize and compare with other different methods.
Response 4: Thank you very much for your suggestions. I also realize that my article lacks some tables to summarize the key experimental data. I have added two tables on lines 369 and 412 of the revised article to summarize and compare the different methods.
Thanks again for taking the time to review my article.
Yours sincerely,
Xusheng Ruan
Reviewer 3 Report
This paper proposed a quasi-bimodal threshold segmentation (QBTS) algorithm and a single-pixel skeleton length measurement (SPSLM) algorithm based on the 8-neighborhood model. The experimental results show that the method proposed in this paper can quickly and accurately segment the target curve from the neon design rendering with complex background interference and measure its length.
The methods and results presented in this paper are meaningful. However, there are maybe some problems in this paper.
1. Some grammar errors shall be corrected and please polish the paper thoroughly. For example, in the Abstract and Conclusion section, please use the past time tense.
2. The presentation of this paper can be improved. And there are some formatting problems.
Author Response
Dear Reviewer:
Thank you very much for your review of my article and your valuable suggestions. After carefully reading your suggestions for revision, I have revised and improved the article. Now I will reply to your questions and suggestions.
Point 1: Some grammar errors shall be corrected and please polish the paper thoroughly. For example, in the Abstract and Conclusion section, please use the past time tense.
Response 1: Thank you very much for your suggestions, which made me realize that there are many areas for improvement in the grammar of my articles. Based on your suggestion, we have re-proofread the article and corrected grammatical errors.
Point 2: The presentation of this paper can be improved. And there are some formatting problems.
Response 2: Thank you very much for your suggestions. I also realize that my article lacks some tables to summarize the key experimental data. I have added two tables on lines 369 and 412 of the revised article to summarize and compare the different methods.
Thanks again for taking the time to review my article.
Yours sincerely,
Xusheng Ruan
Round 2
Reviewer 2 Report
The author has answered my question and revised. It is recommended to publish.